# Effect of Flaxseed Mucilage and Gum Arabic on Probiotic Survival and Quality of Kefir during Cold Storage

**DOI:** 10.3390/foods12030662

**Published:** 2023-02-03

**Authors:** Eiman Alhssan, Songül Şahin Ercan, Hüseyin Bozkurt

**Affiliations:** 1Institute of Sciences, Department of Biochemistry Science and Technology, University of Gaziantep, 27310 Gaziantep, Turkey; 2Department of Food Engineering, Faculty of Engineering, University of Gaziantep, 27310 Gaziantep, Turkey

**Keywords:** kefir, flaxseed, mucilage, probiotic, gum arabic

## Abstract

This study aimed to assess the survival of probiotic cultures in kefir. Kefir is a fermented dairy product, and in this study we incorporated nutritionally rich flaxseed mucilage and gum arabic as a prebiotic, then monitored for improvement in the the viability of *Lactobacillus acidophilus* and *Bifidobacterium lactis.* In addition, some physicochemical variables of kefir were investigated. The addition of flaxseed mucilage and gum arabic significantly (*p* ˂ 0.05) increased the growth of both *Lactobacillus acidophilus* and *Bifidobacterium lactis* compared to the control. Samples enriched with flaxseed mucilage and gum arabic had significantly (*p* ˂ 0.05) reduced pH and increased viscosity. Flaxseed mucilage and gum arabic significantly (*p* ˂ 0.05) changed the color parameters L*, a*, and b*. However, as the concentration of flaxseed mucilage increased, the L* value decreased. Moreover, adding flaxseed mucilage and gum arabic into kefir increased (*p* ˂ 0.05) the protein content. These results showed that flaxseed mucilage and gum arabic could be used to increase the survival of probiotic cultures in kefir without changing its physicochemical properties.

## 1. Introduction

Probiotics have been known for centuries, due to their amazing health benefits. In this context, researchers have regard to probiotic studies relating to fermented milk products. Moreover, consumers prefer to ingest such foods as an alternative to conventional therapies for various chronic diseases. Kefir is one such probiotic product; it is a traditional, acidic, fermented probiotic dairy product that originated in the Caucasus Mountains thousands of years ago. It is made by fermenting any type of pasteurized milk with starter culture or kefir grains [1,2,3]. In addition to beneficial bacteria and yeast, kefir contains vitamins, minerals and essential amino acids which aid the human body in hemostasis, regulation and healing mechanisms [4,5]. Kefir has a wide spectrum of health benefits including physiological, prophylactic and therapeutic properties. These effects are due to the presence of a wide variety of bioactive compounds produced during the fermentation process, and the highly diverse microbial content, which act either independently or synergistically to influence these health benefits [6].

Consumer demand is continuously increasing for natural ingredients that possess specific functional properties and that improve the nutritional value of dairy products. For example, the addition of flaxseed mucilage (FM) and gum arabic (GA). FM is a plant extract obtained from flaxseed; it has a positive effect on the growth of probiotic bacteria. Flaxseed is a member of the *Linaceae* family known as Linseed (*Linum usitatissimum* L.) and it has two basic varieties: brown and yellow [7]. It is considered to be rich in nutrients such as flavonoids, minerals, vitamins, and carbohydrates, which contribute to many potential health benefits. It also contains 20% protein, 7.7% moisture, 3.4% ash, 30–40% oil, as well as alpha-linolenic acid (ALA), which is a precursor of omega-3 fatty acids and which has a positive effect on the growth of *B. lactis* [8]. Moreover, flaxseed contains 35–45% fiber, of which about two-third is insoluble and one third is soluble. This fiber is mucilage [9].

Mucilage is a jelly-like substance that is responsible for holding water in plants, making them drought-resistant. It is extracted by placing seeds into water, and is then filtrated [10]. Flaxseed mucilage can be used as a food gum due to its rheological properties such as thickening, emulsification, and gelling. Mucilage is used as a food additive by food manufactures [11]. Mucilage has many practical uses in the modern world such as: treating burns, wounds, ulcers, irritation, diarrhea, constipation, diabetes and cardiovascular disease; protection against colon cancer; treatment of obesity; and many other beneficial effects [12].

Gum arabic (GA, E-Number 414) is an edible, dried, gummy exudate from the stems and branches of *Acacia senegal* and *A. seyal* that is rich in non-viscous soluble fiber [12]. Many researchers have studied the effect of gum arabic and other types of gums on the growth and activity of probiotic bacteria [13,14,15,16].

Consumers’ interest in healthy diets and wellness has led to an increase in the consumption of foods containing probiotics and prebiotics. Prebiotics are non-digestible carbohydrates that reach the colon, where they are selectively fermented, stimulating the growth and/or activity of one or a limited number of beneficial bacteria. Probiotics and prebiotics, or combinations thereof, are choice in new food developments due to their ability to improve gut health and body comfort. According to previous research, one probable method to enhance the growth and firmness of probiotic bacteria is to fortify dairy products with prebiotics [17].

Researchers used gums and polysaccharides as a prebiotic source, to investigate their effect on the survival of probiotics, and found that they could improve the survival of probiotics. When prebiotics and probiotics are used together, they show symbiotic behavior and provide beneficial effects to the host [18,19]. Also, some researchers observed that flaxseed could increase the viability of probiotics and improve some other properties of the product [20,21]. To the best of our knowledge there have been no studies about the new properties and efficacy of kefir after adding GA and FM in different concentrations in order to determine the symbiotic effect of this combination on *Lactobacillus acidophilus* and *Bifidobacterium lactis* counts within 28 days of storage. Nor have there been any studies to show how FM and GA can affect pH, viscosity, titratable acidity, color, protein and the total solid values of kefir during storage time.

## 2. Materials and Methods

### 2.1. Extraction of Flaxseed Mucilage

About 100 g flaxseed was made up to 2 L with distilled water in a glass beaker and stirred by magnetic stirrer (Ms300Hs, Mtops, Seoul, Republic of Korea) for 3 h (at 55 °C and 60 rpm). Then, the FM was filtrated from the seeds using muslin cloth. The resulting liquid, mucilage, was distributed in small quantities (about 150 mL) into small pyrex glass dishes and put in a forced convection oven (JSOF-100, JS Research Inc. Gongju-City, Republic of Korea) overnight at 55 °C. Then, the dried mucilage (moisture content 3.2%) was collected and stored at 4 °C up to usage.

### 2.2. Experimental Design

Raw cow’s milk was pasteurized at 90 °C for 5 min and cooled down to 25 °C. pH of milk was recorded as 6.22. Then kefir (*Lactobacillus kefiranofaciens*, *Lactobacillus acidophilus*, *Lactobacillus casei*, *Lactobacillus reuteri*, *Lactobacillus plantarum*, *Streptococcus thermophiles*, *Leuconostoc mesenteroides* and at least 1 × 10^8^ log CFU/ mL viable probiotic bacteria) and yogurt starter culture (*Lactobacillus bulgaricus* and *Streptococcus thermophiles*) were inoculated as 0.02% (*w*/*v*), 0.03% (*w*/*v*), respectively. After that, *Bifidobacterium lactis* was added into the mixture as 0.06% (*w*/*v*) after activation at 30 °C for 7 h. The mixture was divided into 9 parts (Table 1). Thereafter, FM was added at a concentration of 0.03, 0.05, and 0.1% (*w*/*v*) where GA was added into each mixture at 0.2% (*w*/*v*). Each part was stirred with magnetic stirrer (Ms300Hs, Mtops, China) for 20 min to dissolve FM and/or GA. Next, each 9 group samples divided into 6 according to each storage time, defined as 0, 1st, 7th, 14th, 21th and 28th days. Then, all samples were put in an incubator (Nüve ES 500, Turkey) at 25 °C for 24 h up to the pH value reached to 4.0. After that, each of the samples was stored at refrigerator at 4 °C for 28 days. All parameters were measured at the sampling time of 0, 1st, 7th, 14th, 21st and 28th days in duplicate.

### 2.3. pH and Acidity

The pH values were recorded using a pH meter (pH/mV/Cond./TDS/Temp. meter 86505), on the 0, 1st, 7th, 14th, 21st and 28th days, at a temperature of 25 °C.

Acidity was measured by adding 3 drops of phenolphthalein to each sample and titrating it with 0.1 NaOH. The % acidity was measured and calculated as lactic acid% [22].
% lactic acid = v × (0.009) × 100/m 
where v is the volume of titrant and m is the weight of the sample.

### 2.4. Viscosity

The viscosity of kefir was measured by a viscometer (Brookfield, DV3T™ viscometer, Worcester County, MA, USA). Viscosity measurement was carried out using V-72(72) spindle of 30 rpm at constant temperature (25 °C). Viscosity was recorded after 25 s and 250 mL sample was used for each run [22]. Viscosity data were expressed in cp.

### 2.5. Color

The colors (L*, a* and b*) of all kefir samples were measured using Hunter lab ColorFlex (A60-1010-615 Model colorimeter, Hunter lab, Reston, VA, USA) on the 0, 1st, 7th, 14th, 21st, and 28th, days of storage [22]. Each time, white and black ceramic plates were used for standardization of the instrument (L_0_ = 93.01, a_0_ = −1.11, and b_0_ = 1.30). The Hunter L, a, and b values correspond to lightness, greenness (−a) or redness (+a), and blueness (−b) or yellowness (+b), respectively. The color measurements were performed at room temperature (25 ± 2 °C) in duplicate.

### 2.6. Microbial Analysis

The enumerations of specific lactic acid bacteria were specified as log CFU/mL. Lactic acid bacteria counts were conducted using de Man, Rogosa and Sharpe (MRS) enriched agar. MRS agar was enriched with 1% maltose and 1% raffinose for *Lactobacillus acidophilus* and *Bifidobacterium lactis* counts, respectively. All kefir samples were diluted tenfold using peptone water (0.2%, *v*/*v*), then spread on MRS enriched agar and incubated at 30 °C and 5% CO_2_ for 72 h (3 days). Then, the numbers of colonies were calculated. Enumerations of bacteria were conducted in duplicate on petri dishes. The counts of bacteria in kefir fortified with 0.2% (*w*/*v*) GA with 0.03, 0.05, 0.1% (*w*/*v*) of FM, 0.03% (*w*/*v*) yoghurt starter culture and 0.06% (*w*/*v*) *Bifidobacterium lactis* were recorded on the 0, 1st, 7th, 14th, 21st and 28th days of storage. The counts of *Lactobacillus acidophilus* and *Bifidobacterium lactis* were recorded according to Dave and Shah [23].

### 2.7. Total Solid and Protein Analysis

Total solid content was quantified using the oven method prescribed by the Turkish Standards Institute (TSI) [22]. Approximately 3 g of kefir (ws) was placed in a pre-weighed (w_1_), pre dried small glass dish and transferred to a hot air oven at 105 °C up to constant weight reached. Samples were cooled in a desiccator before final weights were recorded (w_2_).



$$\mathrm{Total}\;\mathrm{solid}=100-\mathrm{moisture}\%\;\mathrm{Moisture}\%=\frac{\left(w_{2}-\;w_{1}\right)}{\mathrm{ws}}\times 100\%$$



Protein contents of samples were determined according to the TSI [24]. Ten milliliters of sample, 0.5 mL of 0.5% phenolphthalein indicator and 0.4 mL of neutral saturated potassium oxalate were mixed in a conical flask. Then, the mixture was neutralized with 0.1 M NaOH until getting a standard pink color. After that, 2 mL of formalin (37% formaldehyde) was added and titrated again (a). Two milliliters of formalin and 10 mL of water were titrated separately with the same alkali (b) as blank. The protein content of the sample was calculated as follows [24]:% Protein = 1.7 × (a − b)

### 2.8. Sensory Analysis

Hedonic sensory analysis was performed by fifteen trained assessors (graduate students in Gaziantep University Food Engineering Department) to estimate the kefir samples according to the following characteristics: overall appearance (rheology), color, taste/flavor, smell/odor, thickness. Scores were given to each sample as follows: liked-3, normal-2, dislike-1 according to the standard TS EN ISO 8589 [25].

### 2.9. Statistical Analysis

The differences between samples were assessing using one-way ANOVA to compare probiotic bacteria counts, pH, acidity, viscosity and colors under different conditions. Statistical analyses were carried out using the SPSS statistical package version 26.0 (IBM Corporation, Armonk, NY, USA). The Duncan multiple range test was performed to find a statistically significant group at α = 0.05 level. Pearson’s correlation was also applied for correlation analysis. It was used to determine the correlation coefficients between the microbial counts (*L. acidophilus* and *B. lactis*), pH, acidity, viscosity and color parameters (L*, a*, b*).

## 3. Results and Discussion

### 3.1. Viability of Lactobacillus acidophilus

The use of 0.03, 0.05 and 0.1% FM and 0.2% GA caused a significant (*p* ˂ 0.05) increase in *L. acidophilus* compared to control samples (Table 2). The count of viable *L. acidophilus* initiated from 0 day up to 21st days of storage increased compared to control. Samples with 0.03% FM and 0.2% GA had the highest count of *L. acidophilus* (5.95 log CFU/mL) at 21st days. FM is considered to be a prebiotic, and it has a role in improving probiotic growth [26]. Furthermore, the relationship between the counts of *L. acidophilus* and the passing of time was significant according to the Pearson’s test (r = 0.987). Many studies have proven the effect of certain gums (as a source of prebiotics) on the viability and growth of live bacteria [13,14,15,16]. GA is considered to be a prebiotic source, and it has been shown to have a symbiotic effect when it is mixed with probiotics [27]. In another study, the count of *L. acidophilus* bacteria increased as concentrations of GA increased after 21 days of storage [28].

### 3.2. Growth of Bifidobacterium lactis

Changes in the count of viable *Bifidobacterium lactis* in all samples were monitored during storage (Table 3). The addition of 0.03, 0.05 and 0.1% FM alone and in combination with 0.2% GA caused an increase in the growth of *Bifidobacterium lactis* compared with control samples. The highest *B. lactis* count was obtained on the 14th day for the samples enriched with 0.1% FM (7.24 log CFU/mL). Pearson’s test showed that there was a significant positive correlation between the *B. lactis* count and the composition of the sample (r = 0.985). FM is prebiotic matter that contains a mixture of rhamnogalacturonan I and arabinoxylan, which are neutral polysaccharides [29]. These compounds could enhance the survival of *B. lactis*. Another study indicated that flaxseed oil with high level of α-linolenic acid (ALA) has a positive effect on the growth of *Bifidobacterium* [30]. It was reported that GA and Tara gum were good types of gum to increase the count and viability of *B. lactis* [15]. Furthermore, FM that is rich in dietary fibers can be used to form a new prebiotic product for use in the food industry [31,32].

### 3.3. pH Value

Samples enriched with 0.03, 0.05 and 0.1% FM and/or 0.2% GA demonstrated a significant decrease in pH levels across all storage times compared to the control (Figure 1). Some researchers observed that a pH reduction in kefir samples composed of flaxseed and milk correlated with the cell count. In the case of lactobacilli, the pH dropped below 4.0, which is evidence of over fermentation [30]. This result was consistent with the findings of another study, which showed that the pH value decreased in all samples of yoghurt enriched with FM [20]. It was also reported that an increase in acidity correlated with the activity of probiotic bacteria [20]. In this study, the lowest pH value was obtained in samples enriched with 0.2% GA and 0.03% yoghurt culture + 0.06% *Bifidobacterium.* The pH values were 3.95 and 3.92, respectively. Moreover, the counts of viable *L. acidophilus* and *Bifidobacterium* in these samples were less than those of the control. This effect may therefore be caused by post-acidification due to persistent metabolic activity of the product’s micro flora, which decreased lactic acid bacteria count. According to the Pearson’s test, the pH value was negatively correlated with time, *L. acidophilus* and *B. lactis* counts, all color parameters (L*, a*, b*) and viscosity (r = 0.997).

### 3.4. Acidity

The use of 0.03, 0.05 and 0.1% FM and 0.2% GA led to increased acidity on the 0, 1st, 7th, 14th, 21st and 28th days compared with the control (Figure 2). At the termination of refrigerated storage, the samples of kefir that were fortified with FM alone showed increased acidity compared to the kefir samples having no mucilage, establishing that mucilage improves the viability of lactic acid bacteria [26]. Furthermore, the activity of probiotics in mucilage may produce more short-chain fatty acids in dissociated form, causing higher acidity, but this action does not necessarily cause a lower pH value. Pearson’s test indicated that acidity is positively correlated with time, *L. acidophilus* and *B. lactis* counts, viscosity and all color parameters detected in this study (r = 0.984).

### 3.5. The Viscosity

Statistical analysis (ANOVA) indicated that the concentration of FM, GA and yogurt culture with *Bifidobacterium* had a significant effect (*p* ˂ 0.05) on the viscosity value shown in Table 4. The increase in viscosity may be due to the interactions between polysaccharides in FM and the dairy protein [33]. However, the viscosity value decreased as the FM concentration increased at sampling time during storage. This may be due to the increase in the concentration of FM, which caused a decrease in viscosity and the constitution of samples due to protein rearrangement or protein–protein contact. It may also be due to the symbiotic effect of probiotics and prebiotics. Gum tragacanth could increase the viscosity value of Doogh samples, which could be a result of the functional properties of its soluble and insoluble fractions [34]. According to the Pearson’s test, there was a negative correlation between pH and viscosity (*p* ≤ 0.01). Furthermore, there was a positive correlation between viscosity and time, acidity, *B. lactis* count, *L. acidophilus* count, and all color parameters (r = 0.990).

The highest viscosity value was obtained in samples of 0.03% FM during storage. The change in viscosity value may in fact be due to alterations in the volume of casein micelles (supramolecule of colloidal size) during storage, which could be a result of many factors such as the existence and number of fats and proteins, or micelle-binding [35].

### 3.6. Color Parameters (L*, a* and b*)

Hunter color parameters—a* (red-green), b* (yellow-blue), L* (light-dark)—were recorded to show the gradation of visual colors [36]. It is known that whiteness in dairy products is related to colloidal particles such as casein micelles and milk fat globules.

Kefir samples fortified with 0.03, 0.05 and 0.1% FM and 0.2% GA showed an increased (*p* ˂ 0.05) L* value in comparison with control at days 0 and 14 of storage (Figure 3). However, as the concentration of FM increased, the L* value decreased. Similar results were reported in the literature; the L* value of the semi-fat yoghurt decreased as concentrations of FM (*p* < 0.01) increased. It could be that the FM had a darkening effect, possibly due to the absorption of water. The addition of 0.03, 0.05 and 0.1% FM and/or 0.2% GA increased the (*p* ˂ 0.05) a* value compared to the control across all sampling times (Table 5). This result was similar to that of a study which showed that the a* value of yoghurt increased (*p* < 0.01) as the amount of FM increased [37]. The use of 0.03, 0.05 and 0.1% FM and 0.2% GA significantly increased the b* value (yellowness) compared to the control across all days of storage (Figure 4). A similar effect was reported in yogurt samples with FM [37]. The a* and b* values of the yoghurt increased (*p* < 0.01) with the amount of FM, and the samples containing 0.2% FM had the highest a* and b* values among the samples [37]. Pearson’s test showed that there was a significant positive correlation between viscosity and L*, a* and b* values (r = 0.988). In addition, the same test indicated that pH negatively correlated with L*, a* and b* values.

### 3.7. Total Solid and Protein Measurements

The total solid contents of: the control; yoghurt culture and samples containing *Bifidobacterium lactis*; samples with FM and GA added (0.03, 0.05 and 0.1%); and samples containing FM in different concentrations [(0.03, 0.05 and 0.1%) with (0.2% GA), were measured at the termination of storage period. The total solid content of the samples are shown in Table 6. The incorporation of FM and GA into kefir resulted in approximately the same total solid level as the control, and the highest total solid content was recorded in samples containing 0.2% GA.

The protein content of all samples was measured at the end of the storage time. The protein content of the samples is shown in Table 6. All samples with FM and GA had higher protein content than the control. Increasing the concentration of FM increased the protein content, and the highest value was observed in the sample containing 0.1% FM. Moreover, the percentage of protein increased as the GA concentration increased in the yogurt samples [28]. The samples containing Flaxseed had higher protein content than other samples [7].

### 3.8. Sensory Analysis

Enriching the kefir samples with FM had a significant effect on the sensory scores; smell and thickness were highly influenced by the addition of FM and GA, with these samples receiving the lowest scores from the panelists throughout the storage period. Increasing the concentration of FM increased the appearance score, but it was still lower than that of the control. The color scores were approximately the same among samples containing FM (Figure 5).

## 4. Conclusions

In this study, the addition of 0.03, 0.05 and 0.1% FM alone, or mixed with 0.2% GA, into kefir affected the counts of viable *L. acidophilus*, *B. lactis*, pH value, acidity, viscosity, color and protein value on the 0, 1st, 7th, 14th, 21st and 28th days of storage. The FM significantly improved the survival of *L. acidophilus* and *B. lactis*, as well as affecting the other physicochemical properties of the kefir. Therefore, it could be concluded that FM has good potential to improve kefir’s quality in respect of its health benefits. Although it could result in lower sensory scores compared to the control, it is a good option as a complementary food for human health in general, and in particular for people who suffer from gastrointestinal problems and autoimmune weakness.

## Figures and Tables

**Figure 1 foods-12-00662-f001:**
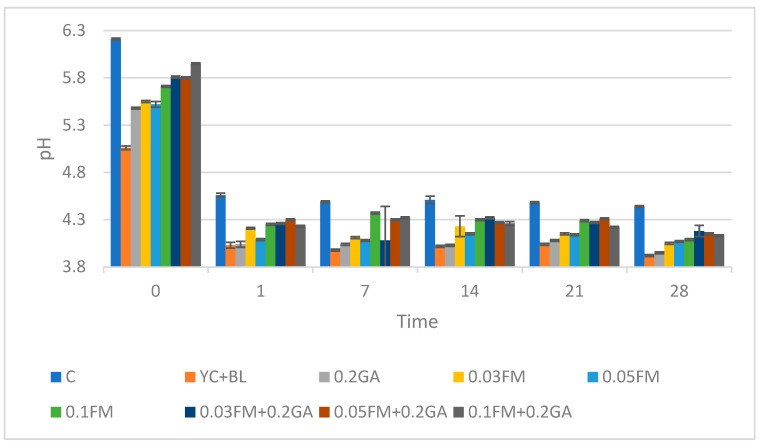
Effect of addition of FM, GA and yogurt culture and B. lactis at different concentrations on pH value in kefir samples. Abbreviations: C, control; YC, yogurt culture; B.L, *B. lactis*; GA, gum arabic; FM, flaxseed mucilage.

**Figure 2 foods-12-00662-f002:**
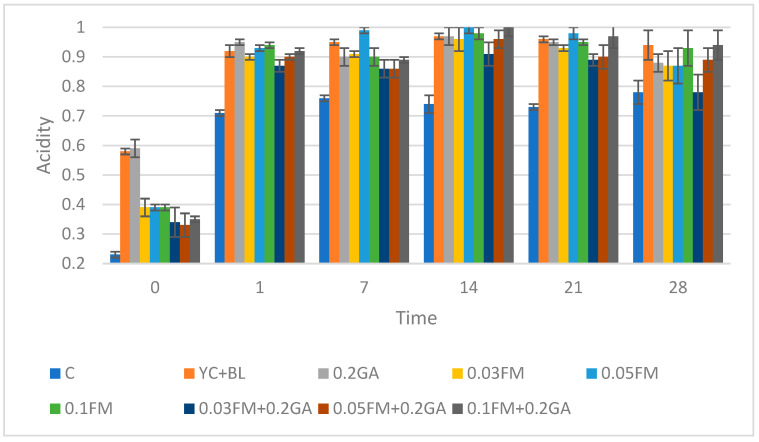
Effect of addition of FM, GA, yogurt culture and *B. lactis* at different concentration on acidity value in kefir samples. Abbreviations: C, control; YC, yogurt culture; B.L, *B. lactis*; GA, gum arabic; FM, flaxseed mucilage.

**Figure 3 foods-12-00662-f003:**
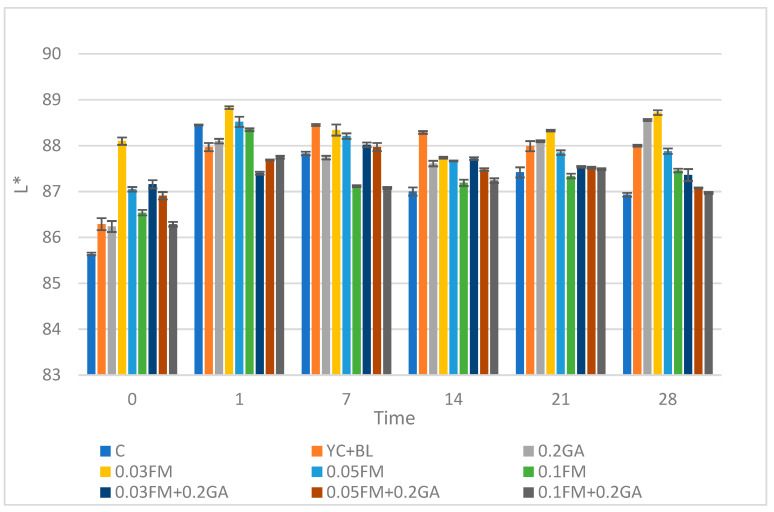
Effect of addition of FM, GA and yogurt culture and *B. lactis* at different concentrations on color L* value in kefir samples during storage time. Abbreviations: C, control; YC, yogurt culture; B.L, *B. lactis*; GA, gum arabic; FM, flaxseed mucilage.

**Figure 4 foods-12-00662-f004:**
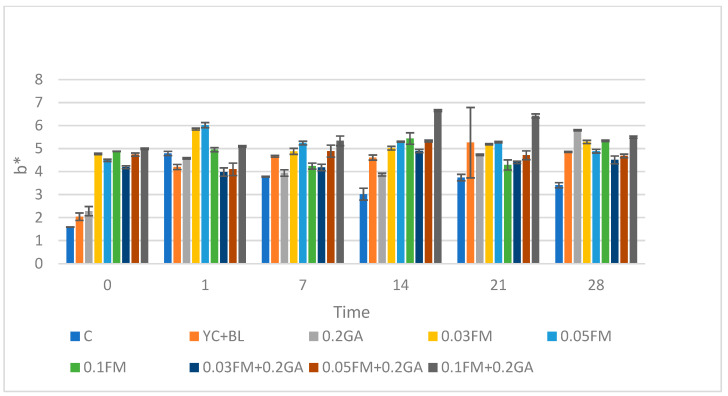
Effect of addition of FM, GA and yogurt culture and *B. lactis* at different concentration on color b* value in kefir samples during storage time. Abbreviations: C, control; YC, yogurt culture; B.L, *B. lactis*; GA, gum arabic; FM, flaxseed mucilage.

**Figure 5 foods-12-00662-f005:**
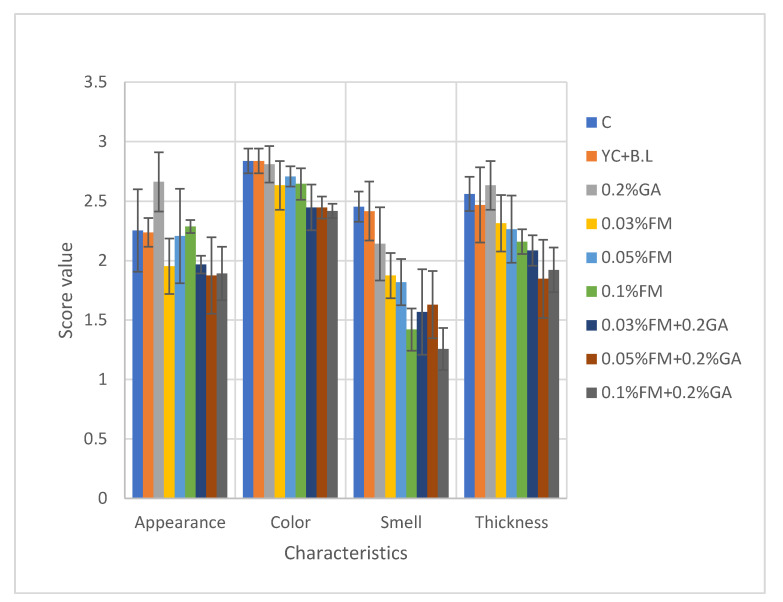
Effect of FM and GA on sensory evaluations during storage time. All scores are the mean of seven data points, n = 31. Abbreviations: C, control; YC, yogurt culture; B.L, *B. lactis*; GA, gum arabic; FM, flaxseed mucilage.

**Table 1 foods-12-00662-t001:** Composition of Kefir Samples.

Sample	To 100 mL Milk
1	0.02 g KC (control)
2	0.02 g KC + 0.03 g YC + 0.06 g *B. lactis*
3	0.02 g KC + 0.03 g YC + 0.06 g *B. lactis* + 0.2 g GA
4	0.02 g KC + 0.03 g YC + 0.06 g *B. lactis* + 0.03 g FM
5	0.02 g KC + 0.03 g YC + 0.06 g *B. lactis* + 0.05 g FM
6	0.02 g KC + 0.03 g YC + 0.06 g *B. lactis* + 0.1 g FM
7	0.02 g KC + 0.03 g YC + 0.06 g *B. lactis* + 0.2 g GA + 0.03 g FM
8	0.02 g KC + 0.03 g YC + 0.06 g *B.* lactis + 0.2 g GA + 0.05 g FM
9	0.02 g KC + 0.03 g YC + 0.06 g *B. lactis* + 0.2 g GA + 0.1 g FM

KC: control (made with kefir culture); YC: yogurt culture; *B. lactis*: *Bifidobacterium lactis*; GA: gum arabic; FM: flaxseed mucilage.

**Table 2 foods-12-00662-t002:** Effect of addition of flaxseed mucilage (FM), gum arabic (GA), yogurt culture (YC) and *B. lactis* (B.L) on *L. acidophilus* count (log CFU/ mL) in kefir samples during storage time.

Time (Day)						
Product	0 Day	1st Day	7th Day	14th Day	21st Day	28th Day
1	2.00 ± 0.01 ^a,A^	3.17 ± 0.01 ^a,B^	3.18 ± 0.01 ^a,B^	3.92 ± 0.04 ^a,C^	4.95 ± 0.06 ^a,D^	6.16 ± 0.01 ^h,E^
2	2.17 ± 0.01 ^a,A^	3.21 ± 0.02 ^b,B^	3.35 ± 0.01 ^ab,C^	4.52 ± 0.07 ^c,D^	5.26 ± 0.05 ^b,E^	5.82 ± 0.01 ^e,F^
3	2.79 ± 0.30 ^b,A^	3.26 ± 0.01 ^c,AB^	4.10 ± 1.09 ^bc,BC^	4.66 ± 0.15 ^c,CD^	5.49 ± 0.01 ^c,D^	5.56 ± 0.01 ^a,D^
4	3.05 ± 0.17 ^bc,A^	3.43 ± 0.01 ^d,B^	4.64 ± 0.01 ^c,C^	4.67 ± 0.01 ^c,C^	5.62 ± 0.01 ^d,D^	5.59 ± 0.01 ^b,D^
5	3.34 ± 0.45 ^cd,A^	3.61 ± 0.01 ^fg,A^	4.65 ± 0.02 ^c,B^	5.72 ± 0.01 ^f,C^	5.82 ± 0.01 ^e,C^	6.11 ± 0.01 ^g,C^
6	3.48 ± 0.01 ^cd,A^	3.51 ± 0.01 ^e,A^	4.61 ± 0.01 ^c,B^	5.34 ± 0.03 ^d,C^	5.62 ± 0.01 ^d,D^	6.11 ± 0.01 ^g,E^
7	3.60 ± 0.14 ^d,A^	3.62 ± 0.01 ^g,A^	4.76 ± 0.01 ^c,B^	5.51 ± 0.01 ^e,C^	5.95 ± 0.01 ^f,D^	6.01 ± 0.01 ^f,D^
8	3.54 ± 0.01 ^d,A^	3.60 ± 0.01 ^fg,B^	4.20 ± 0.02 ^bc,C^	5.37 ± 0.02 ^d,E^	5.22 ± 0.04 ^b,D^	5.63 ± 0.01 ^c,F^
9	3.30 ± 0.01 ^cd,A^	3.59 ± 0.01 ^f,B^	4.58 ± 0.01 ^c,D^	4.38 ± 0.09 ^b,C^	4.96 ± 0.01 ^a,E^	5.65 ± 0.01 ^d,F^

Lower case letters show statistical difference between samples at each time at α = 0.05 level. Capital letters show difference between time at each product at α = 0.05 level. 1: 0.02 g KC (control), 2: 0.02 g KC + 0.03 g YC + 0.06 g *B. lactis*, 3: 0.02 g KC + 0.03 g YC + 0.06 g *B. lactis* + 0.2 g GA, 4: 0.02 g KC + 0.03 g YC + 0.06 g *B. lactis* + 0.03 g FM, 5: 0.02 g KC + 0.03 g YC + 0.06 g *B. lactis* + 0.05 g FM, 6: 0.02 g KC + 0.03 g YC + 0.06 g *B. lactis* + 0.1g FM, 7: 0.02 g KC + 0.03 g YC + 0.06 g *B. lactis* + 0.2 g GA + 0.03 g FM, 80.02 g KC + 0.03 g YC + 0.06 g *B.* lactis + 0.2 g GA + 0.05 g FM, 9: 0.02 g KC + 0.03 g YC + 0.06 g *B. lactis* + 0.2 g GA + 0.1 g FM). Abbreviations: KC: control (made with kefir culture); YC: yogurt culture; *B. lactis*: *Bifidobacterium lactis*; GA: gum arabic; FM: flaxseed mucilage.

**Table 3 foods-12-00662-t003:** Effect of addition of FM, GA and yogurt culture and Bifidobacterium lactis at different concentrations on *B. lactis* count (log CFU/ mL) in kefir samples during storage time.

Time (Day)						
Product	0 Day	1st Day	7th Day	14th Day	21st Day	28th Day
1	5.60 ± 0.01 ^b,A^	6.17 ± 0.01 ^c,B^	6.43 ± 0.01 ^b,C^	6.70 ± 0.00 ^b,E^	6.50 ± 0.01 ^b,D^	6.45 ± 0.03 ^b,C^
2	5.53 ± 0.01 ^a,A^	5.65 ± 0.01 ^a,C^	5.91 ± 0.01 ^a,D^	5.99 ± 0.00 ^a,E^	5.54 ± 0.01 ^a,B^	5.54 ± 0.01 ^a,B^
3	5.82 ± 0.01 ^c,B^	5.87 ± 0.01 ^b,B^	5.88 ± 0.01 ^a,B^	6.05 ± 0.07 ^a,C^	5.47 ± 0.01 ^a,A^	5.43 ± 0.01 ^a,A^
4	6.20 ± 0.01 ^d,A^	6.48 ± 0.01 ^d,B^	6.58 ± 0.00 ^b,c,C^	6.71 ± 0.01 ^b,D^	6.50 ± 0.01 ^b,B^	6.59 ± 0.02 ^b,c,C^
5	6.21 ± 0.00 ^e,A^	6.58 ± 0.02 ^de,C^	6.70 ± 0.01 ^c,D^	6.92 ± 0.01 ^c,E^	6.51 ± 0.01 ^bc,B^	6.61 ± 0.05 ^b,c,C^
6	6.30 ± 0.01 ^f,A^	6.62 ± 0.01 ^e,B^	7.19 ± 0.06 ^d,D^	7.24 ± 0.08 ^d,D^	6.52 ± 0.01 ^bc,B^	6.99 ± 0.16 ^e,C^
7	6.31 ± 0.01 ^f,A^	6.76 ± 0.01 ^f,BC^	6.78 ± 0.01 ^c,BC^	6.91 ± 0.02 ^c,C^	6.67 ± 0.19 ^c,B^	6.89 ± 0.08 ^de,BC^
8	6.39 ± 0.01 ^g,A^	6.80 ± 0.02 ^f,BC^	7.03 ± 0.21 ^d,C^	7.04 ± 0.12 ^c,C^	6.57 ± 0.01 ^bc,AB^	6.76 ± 0.12 ^cd,BC^
9	6.39 ± 0.01 ^g,A^	6.81 ± 0.09 ^f,BC^	7.07 ± 0.17 ^d,D^	7.06 ± 0.09 ^c,D^	6.64 ± 0.01 ^bc,B^	6.94 ± 0.08 ^e,CD^

Lower case letters show statistical difference among samples at each time at α = 0.05 level. Capital letters show difference among time at each product at α = 0.05 level. 1: 0.02 g KC (control), 2: 0.02 g KC + 0.03 g YC + 0.06 g *B. lactis*, 3: 0.02 g KC + 0.03 g YC + 0.06 g *B. lactis* + 0.2 g GA, 4: 0.02 g KC + 0.03 g YC + 0.06 g *B. lactis* + 0.03 g FM, 5: 0.02 g KC + 0.03 g YC + 0.06 g *B. lactis* + 0.05 g FM, 6: 0.02 g KC + 0.03 g YC + 0.06 g *B. lactis* + 0.1 g FM, 7: 0.02 g KC + 0.03 g YC + 0.06 g *B. lactis* + 0.2 g GA + 0.03 g FM, 80.02 g KC + 0.03 g YC + 0.06 g *B.* lactis + 0.2 g GA + 0.05 g FM, 9: 0.02 g KC + 0.03 g YC + 0.06 g *B. lactis* + 0.2 g GA + 0.1 g FM). Abbreviations: KC: control (made with kefir culture); YC: yogurt culture; *B. lactis*: *Bifidobacterium lactis*; GA: gum arabic; FM: flaxseed mucilage.

**Table 4 foods-12-00662-t004:** Effect of addition of FM, GA and yogurt culture and *B. lactis* at different concentration on viscosity value (cp) in kefir samples during storage time.

Time (Day)						
Product	0 Day	1st Day	7th Day	14th Day	21st Day	28th Day
1	13.86 ± 1.29 ^a,A^	1242 ± 83.4 ^a,B^	1623 ± 94.0 ^a,CD^	1738 ± 61.2 ^ab,D^	1554 ± 41.7 ^a,C^	1634 ± 49.5 ^a,CD^
2	369 ± 43.1 ^b,A^	1947 ± 84.9 ^b,B^	2127 ± 74.2 ^c,C^	2375 ± 61.5 ^cd,E^	2166 ± 0.35 ^c,CD^	2308 ± 74.2 ^c,DE^
3	449 ± 30.1 ^bc,A^	2108 ± 129 ^b,BC^	1903 ± 77.4 ^b,B^	2309 ± 99.7 ^c,C^	1911 ± 86.6 ^b,B^	1993 ± 85.6 ^b,B^
4	453 ± 62.8 ^bc,A^	2563 ± 114 ^c,BC^	2807 ± 120 ^f,CD^	2549 ± 120 ^de,BC^	2485 ± 96.5^d,B^	2952 ± 111 ^d,D^
5	502 ± 69.3 ^cd,A^	2100 ± 94.0 ^b,BC^	2377 ± 100 ^d,e,D^	2305 ± 87.7 ^c,CD^	2021 ± 95.5 ^bc,B^	2364 ± 106 ^c,D^
6	689 ± 35.3 ^e,A^	1403 ± 19.4 ^a,B^	1466 ± 30.4 ^a,B^	1637 ± 86.6 ^a,C^	1515 ± 55.2 ^a,BC^	1853 ± 69.3 ^b,D^
7	585 ± 57.6 ^de,A^	2148 ± 118 ^b,C^	2539 ± 61.2 ^e,D,E^	2692 ± 26.2 ^e,E^	2410 ± 79.9 ^d,D^	1969 ± 15.9 ^b,B^
8	830 ± 61.4 ^f,A^	2067 ± 83.4 ^b,B^	2215 ± 110 ^cd,B^	2496 ± 78.5 ^cd,C^	2093 ± 51.3 ^c,B^	2481 ± 104 ^c,C^
9	667 ± 50.7 ^e,A^	1986 ± 79.5 ^b,BC^	2082 ± 72.1 ^bc,C^	1832 ± 78.5 ^b,B^	1861 ± 73.2 ^b,BC^	2351 ± 142 ^c,D^

Lower case letters show statistical difference among samples at each time at α = 0.05 level. Capital letters show difference among time at each product at α = 0.05 level. 1: 0.02 g KC (control), 2: 0.02 g KC + 0.03 g YC + 0.06 g *B. lactis*, 3: 0.02 g KC + 0.03 g YC + 0.06 g *B. lactis* + 0.2 g GA, 4: 0.02 g KC + 0.03 g YC + 0.06 g *B. lactis* + 0.03 g FM, 5: 0.02 g KC + 0.03 g YC + 0.06 g *B. lactis* + 0.05 g FM, 6: 0.02 g KC + 0.03 g YC + 0.06 g *B. lactis* + 0.1 g FM, 7: 0.02 g KC + 0.03 g YC + 0.06 g *B. lactis* + 0.2 g GA + 0.03 g FM, 80.02 g KC + 0.03 g YC + 0.06 g *B.* lactis + 0.2 g GA + 0.05 g FM, 9: 0.02 g KC + 0.03 g YC + 0.06 g *B. lactis* + 0.2 g GA + 0.1 g FM). Abbreviations: KC: control (made with kefir culture); YC: yogurt culture; *B. lactis*: *Bifidobacterium lactis*; GA: gum arabic; FM: flaxseed mucilage.

**Table 5 foods-12-00662-t005:** Effect of addition of FM, GA and yogurt culture and *B. lactis* at different concentration on color a* value in kefir samples during storage time.

Time (Day)						
Product	0 Day	1st Day	7th Day	14th day	21st day	28th day
1	−4.39 ± 0.01 ^a,A^	−3.85 ± 0.05 ^a,B^	−3.88 ± 0.01 ^a,B^	−3.77 ± 0.06 ^a,C^	−3.67 ± 0.02 ^b,D^	−3.63 ± 0.03 ^b,D^
2	−4.23 ± 0.01 ^b,A^	−3.74 ± 0.02 ^b,C^	−3.78 ± 0.02 ^b,B^	−3.78 ± 0.02 ^a,B^	−3.71 ± 0.01 ^b,D^	−3.77 ± 0.01 ^a,BC^
3	−4.13 ± 0.01 ^c,A^	−3.75 ± 0.01 ^b,D^	−3.75 ± 0.00 ^b,D^	−3.75 ± 0.02 ^a,D^	−3.78 ± 0.01 ^a,C^	−3.82 ± 0.01 ^a,B^
4	−3.97 ± 0.01 ^e,A^	−3.24 ± 0.03 ^d,C^	−3.21 ± 0.06 ^d,C^	−3.39 ± 0.03 ^c,B^	−3.24 ± 0.02 ^c,d,C^	−3.18 ± 0.03 ^e,C^
5	−3.97 ± 0.01 ^e,A^	−3.32 ± 0.03 ^c,B^	−3.18 ± 0.01 ^d,D^	−3.27 ± 0.01 ^d,C^	−3.18 ± 0.01 ^d,D^	−3.01 ± 0.03 ^f,E^
6	−3.88 ± 0.01 ^f,A^	−2.87 ± 0.04 ^f,D^	−2.99 ± 0.05 ^ef,C^	−3.45 ± 0.05 ^b,B^	−3.06 ± 0.07 ^e,C^	−2.99 ± 0.01 ^f,C^
7	−3.99 ± 0.02 ^d,A^	−3.15 ± 0.05 ^e,C^	−2.98 ± 0.06 ^f,D^	−3.24 ± 0.06 ^d,B^	−3.06 ± 0.03 ^e,CD^	−3.28 ± 0.07 ^d,B^
8	−3.89 ± 0.01 ^f,A^	−3.13 ± 0.09 ^e,CD^	−3.76 ± 0.06 ^e,CD^	−3.41 ± 0.04 ^bc,B^	−3.03 ± 0.06 ^e,D^	−3.14 ± 0.02 ^e,C^
9	−3.81 ± 0.01 ^g,A^	−3.15 ± 0.02 ^e,C^	−3.37 ± 0.05 ^c,B^	−3.29 ± 0.02 ^d,B^	−3.29 ± 0.02 ^c,B^	−3.39 ± 0.14 ^c,B^

Lower case letters show statistical difference among samples at each time at α = 0.05 level. Capital letters show difference among time at each product at α = 0.05 level. 1: 0.02 g KC (control), 2: 0.02 g KC + 0.03 g YC + 0.06 g *B. lactis*, 3: 0.02 g KC + 0.03 g YC + 0.06 g *B. lactis* + 0.2 g GA, 4: 0.02 g KC + 0.03 g YC + 0.06 g *B. lactis* + 0.03 g FM, 5: 0.02 g KC + 0.03 g YC + 0.06 g *B. lactis* + 0.05 g FM, 6: 0.02 g KC + 0.03 g YC + 0.06 g *B. lactis* + 0.1 g FM, 7: 0.02 g KC + 0.03 g YC + 0.06 g *B. lactis* + 0.2 g GA + 0.03 g FM, 80.02 g KC + 0.03 g YC + 0.06 g *B.* lactis + 0.2 g GA + 0.05 g FM, 9: 0.02 g KC + 0.03 g YC + 0.06 g *B. lactis* + 0.2 g GA + 0.1 g FM). Abbreviations: KC: control (made with kefir culture); YC: yogurt culture; *B. lactis*: *Bifidobacterium lactis*; GA: gum arabic; FM: flaxseed mucilage.

**Table 6 foods-12-00662-t006:** Effect of addition of FM and GA on total solid and protein content after the storage time.

Product	1	2	3	4	5	6	7	8	9
Total solid%	11.63 ± 0.12 ^ab^	11.88 ± 0.17 ^b^	12.05 ± 0.14 ^b^	11.91 ± 0.24 ^b^	11.86 ± 0.04 ^b^	11.75 ± 0.09 ^ab^	11.97 ± 0.36 ^b^	11.74 ± 0.06 ^ab^	11.41 ± 0.16 ^a^
Protein%	12.33 ± 0.60 ^a^	16.15 ± 0.01 ^cd^	16.15 ± 0.60 ^cd^	14.88 ± 0.60 ^c^	15.51 ± 0.90 ^cd^	16.36 ± 0.30 ^d^	13.60 ± 0.00 ^b^	15.08 ± 0.30 ^cd^	15.30 ± 0.60 ^cd^

Lower case letters show statistical difference among samples at each time at α = 0.05 level. 1: 0.02 g KC (control), 2: 0.02 g KC + 0.03 g YC + 0.06 g *B. lactis*, 3: 0.02 g KC + 0.03 g YC + 0.06 g *B. lactis* + 0.2 g GA, 4: 0.02 g KC + 0.03 g YC + 0.06 g *B. lactis* + 0.03 g FM, 5: 0.02 g KC + 0.03 g YC + 0.06 g *B. lactis* + 0.05 g FM, 6: 0.02 g KC + 0.03 g YC + 0.06 g *B. lactis* + 0.1 g FM, 7: 0.02 g KC + 0.03 g YC + 0.06 g *B. lactis* + 0.2 g GA + 0.03 g FM, 80.02 g KC + 0.03 g YC + 0.06 g *B.* lactis + 0.2 g GA + 0.05 g FM, 9: 0.02 g KC + 0.03 g YC + 0.06 g *B. lactis* + 0.2 g GA + 0.1 g FM). Abbreviations: KC: control (made with kefir culture); YC: yogurt culture; *B. lactis*: *Bifidobacterium lactis*; GA: gum arabic; FM: flaxseed mucilage.

## Data Availability

Data is contained within the article.

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
