# Peer review of "Effect of Flaxseed Mucilage and Gum Arabic on Probiotic Survival and Quality of Kefir during Cold Storage"

_foods, 2023, doi:10.3390/foods12030662_

Round 1

Reviewer 1 Report

Please see comments in attached file.

Author Response

January 13, 2023

Thanks for reviewers’ comments concerning our manuscript entitled “EFFECT OF FLAXSEED MUCILAGE AND GUM ARABIC ON KEFIR PROPERTIES” (ID: foods-2158107). It is clear that you have a deep knowledge of the research in the field and ask very specialized questions. Those comments are all valuable and very helpful for revising and improving our paper, as well as the important guiding significance to our researches. We have selected native English speaking professionals to touch up the language and grammar of the text. We have studied the comments carefully and made correction which we hope meet with approval

Yours Sincerely

Dr. Hüseyin Bozkurt (on behalf of the co-authors)

Journal Title              : FOODS

Ms. Ref. No.              : foods-2158107

Manuscript Title   : Effect of flaxseed mucilage and gum arabic on kefir properties

Responses to Editor:

Responses to Reviewer#1

Dear reviewer, thank you very much for your constructive comments, suggestions, and contributions to our manuscript.

Line 2: EFFECT OF FLAXSEED MUCILAGE AND GUM ARABIC ON PROBIOTIC SURVIVAL AND QUALITY OF KEFIR DURING COLD STORAGE

Dear reviewer, thank you very much for your suggestion. We revised the title.

Introduction:

-line 43 - Linseed (Linum usitatissimum L.)

Dear reviewer, thank you very much for your suggestion. We revised it.

-line 45 – ‘too’ changed as ‘to’

Dear reviewer, thank you very much for your suggestion. We revised it.

-line 60 – (Williams &Phillips, 2000) changed as [12].

Dear reviewer, thank you very much for your suggestion. We revised it.

Material and Methods

-line 89 – ‘dried mucilage’ changed as ‘dried mucilage (moisture content 3.2%)’

Dear reviewer, thank you very much for your suggestion. We revised it.

-line 91 – ‘10 min’ changed as ‘5 min’

Dear reviewer, thank you very much for your suggestion. We revised it.

-line 95 – ‘starter culture’ changed as ‘starter culture (Lactobacillus bulgaricus and Streptococcus thermophiles)’

Results and discussion

Dear reviewer, thank you very much for your suggestion. We revised it.

-Table 2- ‘0.00’ was corrected, this correction was done for all tables in manuscript.

Dear reviewer, thank you very much for your suggestion. We revised it.

-Table 2- ‘b,c’ changed as ‘bc’ and all tables in manuscript were changed according to your suggestion.

Dear reviewer, thank you very much for your suggestion. We revised it.

-Table 3-‘f,B,C’ changed as ‘f,BC’ and all tables in manuscript were changed according to your suggestion.

Dear reviewer, thank you very much for your suggestion. We revised it.

-Figure 1- SD was added to figure.

Dear reviewer, thank you very much for your suggestion. We revised it.

-Figure 2- SD was added to figure.

Dear reviewer, thank you very much for your suggestion. We revised it.

-Table 4-‘C,D’ changed as ‘CD’

Dear reviewer, thank you very much for your suggestion. We revised it.

-line 281- results of a* were given in table 5.

Dear reviewer, thank you very much for your suggestion. We revised it.

-line 281- ‘p’ changes as ‘P’ and all manuscript was checked according to the this suggestion.

Dear reviewer, thank you very much for your suggestion. We revised it.

-Figure 3- SD was added to figure.

Dear reviewer, thank you very much for your suggestion. We revised it.

-line 331- changes according to your suggestion.

Dear reviewer, thank you very much for your suggestion. We revised it.

-Figure 4- SD was added to figure.

Dear reviewer, thank you very much for your suggestion. We revised it.

-Table 6- all 3 decimal changed as 2 decimal.

Dear reviewer, thank you very much for your suggestion. We revised it.

-Table 6- ‘0.00’ was corrected.

Dear reviewer, thank you very much for your suggestion. We revised it.

-line 342- at sensory analysis part, you suggested to add a reference for a sentences. But we are sorry that we checked it and could not find the reference for that sentences. So, we have to delete it.

Dear reviewer, thank you very much for your suggestion. We revised it.

-Figure 5- SD was added to figure.

Dear reviewer, thank you very much for your suggestion. We revised it.

We appreciate for Editors/Reviewers’ warm work earnestly, and hope that the correction will meet with approval.

Once again, thank you very much for your comments and suggestions

Best regards for you.

Sincerely yours,

Authors

Reviewer 2 Report

foods-2158107-peer-review-v1

The study investigated the survival probiotic cultures in kefir, focusing on the viability of Lactobacillus acidophilus, Bifidobacterium lactis. This was achieved by adding in the fermentation procedure flaxseed mucilage and gum arabic.  Through their tests the author provided evidence that flaxseed mucilage and gum Arabic combination increase the growth of two important probiotics for human health the Lactobacillus acidophilus and the Bifidobacterium lactis.  

Overall, the paper is interesting but not well written at all. The hypothesis and purpose of study are not clearly and nor concisely presented. Yogurt starter culture use in the experimental procedure should be introduced. In materials and methods section all research components are not present and nor clearly stated.

Abstract should be like this. Please do not use abbreviation in the abstract. There are several grammatical errors along the text that must be resolved.

Abstract

This study aimed to assess the survive of probiotic cultures in kefir. Kefir is a fermented dairy product and here, incorporating nutritionally rich flaxseed mucilage and gum arabic as a prebiotic we monitored the viability of Lactobacillus acidophilus and Bifidobacterium lactis.  Also, some physicochemical variables of kefir were investigated. The addition of flaxseed mucilage and gum arabic significantly (PË‚0.05) increased the growth of both, Lactobacillus acidophilus and Bifidobacterium lactis compared to the control. Samples enriched with flaxseed mucilage and gum arabic had significantly (PË‚0.05) decreased the pH and increased viscosity. Flaxseed mucilage and gum arabic significantly (PË‚0.05) changed the color parameters L*, a*, and b*. However, as the concentration of flaxseed mucilage increased, the L* value decreased. Moreover, adding flaxseed mucilage and gum arabic into kefir increased (PË‚0.05) the protein contents. These results showed that flaxseed mucilage and gum arabic could use to increase the survival of probiotic cultures in kefir without changing its physicochemical properties.

Author Response

January 16, 2023

Thanks for reviewers’ comments concerning our manuscript entitled “EFFECT OF FLAXSEED MUCILAGE AND GUM ARABIC ON KEFIR PROPERTIES” (ID: foods-2158107). It is clear that you have a deep knowledge of the research in the field and ask very specialized questions. Those comments are all valuable and very helpful for revising and improving our paper, as well as the important guiding significance to our researches. We have selected native English speaking professionals to touch up the language and grammar of the text. We have studied the comments carefully and made correction which we hope meet with approval

Yours Sincerely

Dr. Hüseyin Bozkurt (on behalf of the co-authors)

Journal Title            : FOODS

Ms. Ref. No.              : foods-2158107

Manuscript Title   : Effect of flaxseed mucilage and gum arabic on kefir properties

Responses to Editor:

Responses to Reviewer#2

The study investigated the survival probiotic cultures in kefir, focusing on the viability of Lactobacillus acidophilus, Bifidobacterium lactis. This was achieved by adding in the fermentation procedure flaxseed mucilage and gum arabic.  Through their tests the author provided evidence that flaxseed mucilage and gum Arabic combination increase the growth of two important probiotics for human health the Lactobacillus acidophilus and the Bifidobacterium lactis.  

Overall, the paper is interesting but not well written at all. The hypothesis and purpose of study are not clearly and nor concisely presented. Yogurt starter culture use in the experimental procedure should be introduced. In materials and methods section all research components are not present and nor clearly stated.

Abstract should be like this. Please do not use abbreviation in the abstract. There are several grammatical errors along the text that must be resolved.

Abstract

This study aimed to assess the survive of probiotic cultures in kefir. Kefir is a fermented dairy product and here, incorporating nutritionally rich flaxseed mucilage and gum arabic as a prebiotic we monitored the viability of Lactobacillus acidophilus and Bifidobacterium lactis.  Also, some physicochemical variables of kefir were investigated. The addition of flaxseed mucilage and gum arabic significantly (PË‚0.05) increased the growth of both, Lactobacillus acidophilus and Bifidobacterium lactis compared to the control. Samples enriched with flaxseed mucilage and gum arabic had significantly (PË‚0.05) decreased the pH and increased viscosity. Flaxseed mucilage and gum arabic significantly (PË‚0.05) changed the color parameters L*, a*, and b*. However, as the concentration of flaxseed mucilage increased, the L* value decreased. Moreover, adding flaxseed mucilage and gum arabic into kefir increased (PË‚0.05) the protein contents. These results showed that flaxseed mucilage and gum arabic could use to increase the survival of probiotic cultures in kefir without changing its physicochemical properties.

Dear reviewer, thank you very much for your constructive comments, suggestions, and contributions to our manuscript

Abstract:

- Dear reviewer, thank you for your suggestion. We revised the abstract according to the suggestion.

- The hypothesis and purpose of study are not clearly.

- The hypothesis and purpose were checked and rewritten again according to your suggestion.

Dear reviewer, thank you very much for your suggestion. We revised it

- Yogurt starter culture use in the experimental procedure should be introduced.

- Yogurt starter culture use in the experimental procedure was added.

Dear reviewer, thank you very much for your suggestion. We revised it.

- In materials and methods section all research components are not present and nor clearly stated.

Dear reviewer, thank you very much for your suggestion. We checked materials and methods section all research components revised it.

- There are several grammatical errors along the text that must be resolved.

Dear reviewer, thank you for your suggestion. We checked all text, all grammatical errors were corrected.

We appreciate for Editors/Reviewers’ warm work earnestly, and hope that the correction will meet with approval.

Once again, thank you very much for your comments and suggestions

Best regards for you.

Sincerely yours,

Authors

Round 2

Reviewer 2 Report

foods-2158107-peer-review-v2

Dear authors, this revised form of the manuscript, improved substantially in the presentation of results, as I said before of interest in the panorama of kefir beverage enriched by fiber. 

Minor

lines 43 and 44 take out the words "means “very useful”"

at lines 71 and 72 add these reference as a good examples  10.3390/molecules25112706 10.3390/molecules26206187

line 134 define MRS as de Man, Rogosa and Sharpe (MRS) agar 

Author Response

January 19, 2023

Dear Editor,

Thanks for reviewers’ comments concerning our manuscript entitled “EFFECT OF FLAXSEED MUCILAGE AND GUM ARABIC ON KEFIR PROPERTIES” (ID: foods-2158107). It is clear that you have a deep knowledge of the research in the field and ask very specialized questions. Those comments are all valuable and very helpful for revising and improving our paper, as well as the important guiding significance to our researches. We have studied the comments carefully and made correction which we hope meet with approval

Yours Sincerely

Dr. Hüseyin Bozkurt (on behalf of the co-authors)

Journal Title              : FOODS

Ms. Ref. No.              : foods-2158107

Manuscript Title   : Effect of flaxseed mucilage and gum arabic on kefir properties

Responses to Editor:

Responses to Reviewer#2

Dear reviewer, thank you very much for your constructive comments, suggestions, and contributions to our manuscript.

- lines 43 and 44 take out the words "means “very useful”

We revised it as ‘Flaxseed is a member of Linaceae family, known as Linseed (Linum usitatissimum L.) and it has two basic varieties: brown and yellow’.

Dear reviewer, thank you very much for your suggestion.

- at lines 71 and 72 add these reference as a good examples  10.3390/molecules25112706 10.3390/molecules26206187

We revised it as ‘When prebiotics and probiotics come together, they show symbiotic behavior and provide beneficial effects to the host [18, 19].’

Also, all references in manuscript were rearranged according to your suggestion. References of  [18, 19] were added to references part of manuscript.

Dear reviewer, thank you very much for your suggestion.

- line 134 MRS corrected as de Man, Rogosa and Sharpe (MRS) agar 

Dear reviewer, thank you very much for your suggestion. We revised it.

  • All corrections were painted to blue color.

We appreciate for Editors/Reviewers’ warm work earnestly, and hope that the correction will meet with approval.

Once again, thank you very much for your comments and suggestions

Best regards for you.

Sincerely yours,

Authors
